# Living Mulches, Rolled Cover Crops, and Plastic Mulch: Effects on Soil Properties, Weed Suppression, and Yield in Organic Strawberry Systems

**DOI:** 10.3390/plants14213385

**Published:** 2025-11-05

**Authors:** Arianna Bozzolo, Jacob Pecenka, Andrew Smith

**Affiliations:** 1Rodale Institute, California Organic Center, 1012 W Ventura Blvd., Camarillo, CA 93010, USA; jacob.pecenka@rodaleinstitute.org; 2Rodale Institute, 611 Siegfriedale Road, Kutztown, PA 19530, USA

**Keywords:** organic strawberries (*Fragaria × ananassa* ‘Albion’), cover crop termination, roller crimper, soil health, weed suppression, plastic mulch alternatives, microplastic pollution

## Abstract

Plastic mulch is widely used in organic strawberry production but raises sustainability concerns due to its persistence, disposal challenges, and contribution to microplastic pollution. This study evaluated the potential of high-residue cover crops and living mulches as alternatives to plastic mulch in coastal California. Over two seasons (2022–2024), we compared five mulching treatments: black polyethylene mulch (Plastic); a white clover (*Trifolium repens*) living mulch (Clover); two roller-crimped sorghum–sudangrass and field pea mixtures (Sorghum 1, Sorghum 2); and a roller-crimped buckwheat–pea mixture (Buckwheat). The objectives were to evaluate the effectiveness of these treatments on (i) soil properties and biological indicators, (ii) weed suppression, and (iii) strawberry yield in organic systems. A schematic timeline was developed to depict cover-crop growth, termination, and strawberry production across both years. Compost (10 t·ha^−1^) and fish emulsion (5–1–1 NPK, 4 L·ha^−1^ biweekly) were applied to all treatments during fruiting. Sorghum residues produced the highest biomass (up to 23 t·ha^−1^) and supported yields comparable to plastic mulch in 2023. Under lower-yield conditions in 2024, sorghum-based treatments outperformed plastic. Soil responses were modest and time-point specific: Sorghum 1 showed higher organic C and organic N pre-harvest in 2023, and both sorghum treatments increased soil organic matter pre-harvest in 2024. Biological indicators such as CO_2_–C and microbially active carbon declined seasonally across all treatments, indicating strong temporal control. Weed outcomes diverged by system—Clover suppressed weeds effectively but reduced yield by >50% due to competition, while Buckwheat decomposed rapidly and provided limited late-season suppression. These results demonstrate that rolled high-residue cover crops, particularly sorghum-based systems, can reduce dependence on plastic mulch while maintaining yields and enhancing soil cover. Living mulches and short-lived covers may complement residue systems when managed to minimize competition and extend ground cover.

## 1. Introduction

Polyethylene mulch is widely used in commercial strawberry (*Fragaria × ananassa*) production to suppress weeds, conserve soil moisture, and increase early yields by warming the soil. However, its extensive use has raised growing concerns about long-term sustainability. Recent studies show that plastic residues can persist in soils for decades, contributing to soil degradation, microplastic accumulation, and potential food-chain contamination. These issues have sparked regulatory scrutiny in regions such as California, where agricultural plastic use is under review, and have intensified efforts to identify alternatives that maintain productivity while reducing environmental impact [1,2,3,4,5,6].

Among the most promising strategies are biomass-based mulching systems, including high-residue cover crops terminated with a roller crimper (“rolled residues”) and living mulches. Rolled residues can provide season-long ground cover, suppress weeds by shading and physical impedance, and buffer soil temperature and moisture. Their effectiveness depends on biomass quantity, species composition, termination efficacy, and residue persistence [7,8,9]. Warm-season grasses such as sorghum–sudangrass produce abundant lignified biomass that persists longer than low-lignin species but can be difficult to terminate completely, often requiring multiple roller passes. Broadleaf species such as buckwheat generate rapid early biomass but decompose quickly, shortening weed suppression [10].

Living mulches, by contrast, provide continuous soil cover and can reduce erosion and weed emergence while improving habitat for beneficial organisms. However, in high-value crops such as strawberries, they can compete with the cash crop for light, water, and nutrients unless carefully managed through mowing, strip-seeding, or temporal staggering [11,12].

Beyond yield and weed control, these alternative mulching systems can also influence soil biological activity and nutrient cycling. Cover crops have been shown to increase microbial activity and soil organic carbon [13,14], though effects can be modest over short timeframes and strongly dependent on local climate and soil type. In contrast, plastic mulch modifies the soil microclimate but contributes no organic inputs and may even alter microbial communities through physical isolation of the soil surface.

Despite increasing interest in plastic-free production systems, few studies have simultaneously compared rolled cover crops and living mulches against conventional plastic mulch under coastal California conditions. The cool, humid spring climate and sandy loam soils of this region influence residue persistence, moisture dynamics, and weed pressure differently from the inland valleys where most research has been conducted. Addressing these site-specific challenges is essential to guide growers in transitioning toward regenerative organic systems that minimize plastic dependence.

Therefore, this study was designed to evaluate and compare rolled cover crop residues and a white clover living mulch with polyethylene mulch in organic strawberry systems. The objectives were to assess how these mulching treatments affect (i) soil properties and biological indicators, (ii) weed suppression, and (iii) strawberry yield.

To guide the evaluation, the following hypotheses were tested:

**H1.** 
*Rolled sorghum–pea residues would maintain yields comparable to plastic while enhancing soil properties.*


**H2.** 
*White clover would strongly suppress weeds but reduce yield due to in-row competition.*


**H3.** 
*Buckwheat residues would provide rapid early weed suppression but less persistence later in the season.*


**H4.** 
*Seasonal dynamics (spring to summer) would exert stronger influence on biological soil indicators (CO_2_–C and MAC) than the mulching treatments themselves.*


Together, these hypotheses address the broader question of whether high-residue cover crop systems can provide agronomic and environmental performance comparable to plastic mulch while advancing the transition toward regenerative organic production in California strawberry systems. This study provides a multi-season comparison of polyethylene mulch, rolled cover crop residues, and a living mulch in organic strawberry production under coastal California conditions, linking yield performance with soil health, weed management, and operational feasibility.

## 2. Results

### 2.1. Soil Chemical and Biological Properties

In 2023 pre-harvest, soil organic matter (SOM) did not differ significantly among treatments (Table 1A). However, organic carbon (OC) and organic nitrogen (Org. N) were significantly higher under Sorghum 1 compared with Buckwheat and Plastic (*p* ≤ 0.05), showing 43% greater OC and 23% greater Org. N than Plastic. Sorghum 2 and Clover showed intermediate values. Total N, CO_2_–C, and microbially active carbon (MAC) did not vary significantly among treatments at this stage.

By the post-harvest 2023 sampling, no significant differences were detected among treatments for any measured soil parameter (Table 1B). Across all treatments, CO_2_–C declined sharply, from roughly 40–45 ppm before harvest to about 9–11 ppm after harvest, indicating reduced microbial respiration following crop removal. MAC values remained variable, suggesting heterogeneous microbial activity across plots.

Unusually high variability in Total N and MAC under Clover and Sorghum 1 post-harvest was likely biological rather than analytical in origin, reflecting localized “hot spots” of organic-matter decomposition and microbial activity near dense root zones.

In 2024 pre-harvest (Table 2A), soil organic matter (SOM) differed significantly among treatments (*p* ≤ 0.05), with Sorghum 1 and Sorghum 2 showing higher SOM than Plastic and Buckwheat, while Clover was intermediate. Organic carbon (OC) and Total N showed moderate temporal variation but no treatment effects.

By the post-harvest 2024 sampling (Table 2B), no significant differences were observed among treatments for any soil parameter except SOM, which remained slightly higher under Sorghum 1. Across treatments, OC and microbially active carbon (MAC) decreased compared with pre-harvest values, consistent with residue decomposition and nutrient uptake during late-season warming.

### 2.2. Soil Infiltration, Moisture, and Temperature

Infiltration rates varied significantly across treatments (Figure 1). Cover-based treatments, particularly Buckwheat, exhibited the highest infiltration rates (*p* ≤ 0.05), indicating improved water movement through the soil profile. Clover and Sorghum 1 followed with intermediate infiltration rates. Plastic and Sorghum 2 had the lowest infiltration, with Plastic significantly lower than Buckwheat and Clover.

Volumetric soil moisture at 10 cm depth differed significantly across treatments during most of the 2023 and 2024 growing seasons (Table 3). Plastic mulch maintained consistently higher soil moisture (by 19–24%) compared with other treatments, particularly from March to June. Sorghum 1 and Sorghum 2 had intermediate moisture retention, while Clover and Buckwheat exhibited the lowest levels, particularly during summer.

Soil temperature showed less variation overall. In both years, plastic mulch slightly increased soil temperature in March and early spring (*p* ≤ 0.001), while other treatments showed no significant differences during midseason. These findings reflect the microclimatic effect of plastic mulch in warming soils early in the growing season.

### 2.3. Cover Crop and Weed Biomass

Aboveground biomass production differed significantly among treatments (Figure 2). In both years, Buckwheat produced the greatest cover biomass (3.2 ± 0.3 t·ha^−1^ in 2023), about four times that of Clover. Sorghum 1 and Sorghum 2 were intermediate and did not differ significantly from each other.

Weed biomass also varied by treatment (Figure 3). Clover maintained the lowest weed biomass at nearly all sampling dates, Plastic and Sorghum 1/2 were intermediate, and Buckwheat had the highest late-season weed growth due to rapid residue decomposition. Weed suppression was not directly proportional to residue mass; rather, living Clover maintained weed control through continuous coverage, whereas Buckwheat’s rapid decline in residue cover reduced suppression after early summer.

### 2.4. Strawberry Yield

Strawberry yields varied significantly among treatments in both years (Table 4).

In 2023, Sorghum 2 (24.5 t·ha^−1^), Sorghum 1 (22.2 t·ha^−1^), and Plastic (23.6 t·ha^−1^) produced comparable total yields and outperformed Buckwheat (13.3 t·ha^−1^). Clover yielded lowest overall (0.6 t·ha^−1^; *p* < 0.001). Plastic produced an earlier April flush (4.2 t·ha^−1^; *p* < 0.001), more than double the early yield of the other treatments.

In 2024, total yields declined across all treatments, reflecting less favorable environmental conditions. Sorghum 1 (11.6 t·ha^−1^) and Sorghum 2 (11.1 t·ha^−1^) had the highest total yields, followed by Buckwheat (5.0 t·ha^−1^), Plastic (2.4 t·ha^−1^), and Clover (0.4 t·ha^−1^).

The large year-to-year yield variation corresponds with lower early-season temperatures and two short heat events during flowering in 2024, which likely limited fruit set. The plastic mulch system, which warms soils, performed worse under these conditions, possibly due to excessive root-zone temperature.

## 3. Discussion

This study compared the performance of high-residue rolled cover crops, a living mulch, and polyethylene mulch in organic strawberry systems across two growing seasons under coastal California conditions. Overall, the results demonstrate that high-residue sorghum–pea systems can achieve comparable or superior yields to plastic mulch while improving soil cover and infiltration, although soil biological responses were modest and time-dependent. The discussion below follows the main study hypotheses (H1–H4) to provide a systematic interpretation of the findings.

Consistent with H1, treatment effects on soil properties were minor and time-specific, as expected for short-term experiments, yet sorghum-based systems showed early signs of improvement. Pre-harvest in 2023, Sorghum 1 showed higher organic carbon and nitrogen, while pre-harvest 2024 revealed modest increases in soil organic matter in both sorghum treatments. These results suggest that high-residue sorghum mixtures can enhance soil carbon pools within one to two years, primarily due to greater biomass inputs and slower decomposition relative to other covers.

The post-harvest increases in total nitrogen observed in 2023, particularly in Clover and Sorghum 1, were attributed to localized nitrogen mineralization and root turnover rather than analytical anomalies. High microbial activity near decomposing root and residue zones may have caused temporary nutrient accumulation. Such localized nutrient “hot spots” are typical in organic systems with high belowground biomass, supporting the idea that soil improvements begin as spatially variable processes before becoming detectable at the field scale. In support of H4, the consistent decline in CO_2_–C from pre- to post-harvest across all treatments reflected reduced microbial respiration due to lower soil moisture and substrate availability during the dry season. This seasonal control was stronger than the treatment effect itself, indicating that temporal factors currently overshadow short-term management differences.

These temporal patterns—rising N pools followed by declining CO_2_–C—are consistent with short-term mineralization and the seasonality typical of Mediterranean climates. The loss-on-ignition (LOI) method used for soil organic matter was appropriate for relative comparisons but may slightly overestimate values in alkaline soils (pH ≈ 7.8). Overall, the results support H1 and H4, indicating that while high-residue cover crops initiate measurable shifts in carbon and nitrogen cycling, the influence of seasonal dynamics remains predominant. Future work should include cross-validation with dry combustion to improve quantitative accuracy.

### 3.1. Soil Moisture, Temperature, and Infiltration (H1)

Plastic mulch consistently maintained higher soil moisture, confirming its known role in reducing evaporation and buffering short-term water loss. However, consistent with H1, sorghum-based residues provided intermediate moisture retention while markedly improving infiltration, indicating better water entry and soil structure under residue cover. The enhanced infiltration under cover-based treatments, particularly Buckwheat, suggests that surface residue improved porosity and water movement through the soil profile while reducing crusting. Clover and Sorghum 1 followed with intermediate infiltration rates, whereas Plastic and Sorghum 2 showed the lowest infiltration values.

In both years, plastic slightly increased early-season soil temperature (March–April), enhancing early growth and fruiting. This warming effect, while beneficial for early yield flushes, can also lead to excessive heat accumulation under warmer spring conditions. In contrast, the sorghum-based systems buffered soil temperature fluctuations, providing a more stable microclimate for root growth and microbial activity. The contrasting yield responses between 2023 and 2024 support this interpretation: during the cooler 2023 season, plastic favored early production, whereas in 2024, the moderated temperature under sorghum residues corresponded with higher total yield. These findings indicate that high-residue cover crops not only conserve water but also enhance climate resilience by preventing heat stress and maintaining moisture availability under variable environmental conditions.

Together, these results support H1 by demonstrating that sorghum-based rolled residue systems can moderate soil microclimate—retaining sufficient soil moisture, enhancing infiltration, and stabilizing temperature—thereby maintaining productivity while reducing dependence on plastic mulch.

### 3.2. Weed Suppression (H2, H3)

Weed suppression varied strongly by treatment. Consistent with H2, Clover maintained the lowest weed biomass, demonstrating the competitive potential of continuous ground cover. However, this same competitiveness reduced strawberry yield by more than 50%, highlighting the trade-off between weed suppression and crop competition in living-mulch systems. These results confirm that while living mulches can be effective for weed control, their management must be carefully adjusted—through mowing, strip-planting, or delayed establishment—to reduce in-row competition in high-value crops such as strawberries.

Consistent with H3, buckwheat provided effective early weed control but its rapid residue decomposition diminished suppression later in the season, consistent with previous findings [10]. This rapid decline in cover emphasizes that weed management is governed not only by total biomass but by residue persistence and decomposition rate. Sorghum residues provided moderate, sustained suppression while supporting near-plastic yields, suggesting a balanced performance under organic management. Their coarse, lignified structure contributed to prolonged surface coverage, maintaining partial light interception and physical impedance even as finer residues decomposed.

Overall, these results support both H2 and H3 by demonstrating that continuous living cover achieves the greatest weed suppression but at the expense of crop yield, whereas rolled sorghum residues achieve a compromise between suppression, residue longevity, and productivity. The effectiveness of weed control thus depends on the duration and quality of ground cover rather than residue quantity alone.

### 3.3. Yield Performance and Year-to-Year Variability (H1, H4)

Yield patterns reflected both treatment effects and environmental variability. In line with H1, in 2023, plastic mulch and sorghum systems achieved similar total yields, with plastic producing an earlier harvest flush due to soil warming. This indicates that high-residue sorghum residues maintained comparable productivity despite cooler soil temperatures. In 2024, when cooler early-season temperatures and heat stress events reduced fruit set, the sorghum systems maintained higher yields—likely due to moderated soil temperatures, improved water retention, and gradual nutrient release. These patterns align with H4, highlighting that seasonal environmental conditions exerted a stronger influence on yield outcomes than short-term treatment differences.

The decline in plastic mulch performance in 2024 may also relate to excessive heat accumulation in the root zone, aligning with reports of reduced fruit set under high temperature stress [15]. Thus, the results suggest that sorghum-based systems provide a yield advantage under variable climatic conditions by buffering soil temperature and moisture fluctuations. Sorghum’s resilience under variable climatic conditions supports its potential as a sustainable, climate-adaptive alternative to polyethylene mulch.

### 3.4. Operational and Practical Considerations

A practical limitation of the sorghum-based systems was incomplete termination after the first roller-crimper pass, requiring manual clipping to achieve uniform mulch coverage. This issue is consistent with reports that sorghum–sudangrass can be difficult to terminate effectively when biomass is high or when rolling occurs before full flowering. Future studies should refine termination timing and explore multi-pass or mechanical-assist strategies to improve uniformity and reduce labor costs.

Clover’s strong weed control but poor yield performance suggests it may be better suited as an alley or perimeter cover rather than an in-row mulch. Modifications such as mowed strips or banded living-mulch designs could reduce competition while maintaining soil cover. Buckwheat may serve well as a short-duration or rotational cover preceding strawberries but is unsuitable for season-long suppression without supplemental management.

Because the experiment was conducted on certified organic land, the use of herbicides or other synthetic weed controls typical of conventional bare-soil systems was not permitted. Consequently, a true bare-soil conventional control could not be included. Moreover, unmulched organic beds are rarely maintained in coastal California due to severe weed pressure and erosion risk, so the treatments tested here reflect realistic organic management options.

### 3.5. Broader Implications

The environmental costs of polyethylene mulch—including landfill accumulation, on-farm residue, and microplastic pollution—underscore the urgency of developing biodegradable or regenerative alternatives. Studies have documented microplastic transfer from soil to edible plant tissues, including grains and leafy vegetables, raising concerns about potential food-chain exposure [6]. By substituting plastic with high-residue cover crops, growers can maintain productivity while reducing plastic-derived contaminants.

Although biological and chemical soil improvements occur gradually, transitioning away from plastic mulch provides immediate environmental benefits and supports long-term soil resilience. Broad adoption will depend on optimizing residue termination methods, developing region-specific management practices, and incentivizing research and implementation through programs such as the California Healthy Soils Program and USDA Climate-Smart Agriculture initiatives.

Together, these results support the study hypotheses and demonstrate that biologically based residue systems can meet productivity and soil health goals while reducing dependence on plastic mulch.

## 4. Materials and Methods

### 4.1. Site Description

The experiment was conducted from March 2022 to July 2024 at the Rodale Institute California Organic Center in Camarillo, California (34.2205° N, 119.1082° W). The site is characterized by a warm-summer Mediterranean climate (Köppen classification Csa) with mild winters and dry summers. Average annual precipitation is approximately 380 mm, primarily occurring between November and March (see Table 5), and mean monthly temperatures range from 10 °C in winter to 25 °C in summer.

The soil is a loam to sandy loam, with an average texture of 47% sand, 34% silt, and 19% clay. Baseline soil characteristics prior to experiment initiation included 0.85% organic carbon, 2.1% organic matter, and a pH of 7.84. Additional baseline nutrient values were: phosphorus (32.2 ppm), potassium (283.3 ppm), calcium (2641.3 ppm), and magnesium (532.3 ppm). The cation exchange capacity was 19.2 meq/100 g.

### 4.2. Experimental Design and Treatments

The trial followed standard raised-bed practices for organic strawberry production in coastal California. Beds were 30 cm high and 132 cm apart (center-to-center), each supporting two rows of ‘Albion’ strawberry plants spaced 35 cm between rows and 30 cm in-row. This configuration resulted in a planting density of approximately 50,500 plants per hectare.

The experiment was a randomized complete block design with four replications. Each plot consisted of three raised beds (9 m length, total area 11.88 m^2^) and included sixty strawberry crowns per plot (two rows of 30 plants each). Five mulching treatments were evaluated:

Sorghum 1: Roller-crimped sorghum–sudangrass (*Sorghum bicolor* × *S. sudanense*) + field pea (*Pisum sativum* subsp. *arvense*), 100 kg·ha^−1^ and 30 kg·ha^−1^ seeding rates, respectively.

Sorghum 2: Same mixture but with 200 kg·ha^−1^ sorghum and 30 kg·ha^−1^ pea.

Buckwheat: Roller-crimped buckwheat (*Fagopyrum esculentum*) + field pea, 67 and 30 kg·ha^−1^.

Clover: White clover (*Trifolium repens* L.) living mulch seeded between strawberry rows at transplanting.

Plastic: Black polyethylene mulch applied at transplanting (control).

Cover crops were established in May 2022 and terminated with a roller crimper in August 2022, approximately 90 days after seeding. A schematic timeline (Figure 3) illustrates the sequence of cover-crop growth, termination, and strawberry production across both years. Roller-crimp (I & J Manufacturing, Gap, PA, USA) termination was incomplete in sorghum plots due to biomass density and stem rigidity; regrowth was manually clipped within seven days post-termination and residues redistributed evenly on the bed surface.

At strawberry transplanting (August 2022), 10 t·ha^−1^ of compost (dry weight basis) was incorporated, and organic fish emulsion (5–1–1 NPK) was fertigated biweekly at 4 L·ha^−1^ during active fruiting. No synthetic fertilizers or pesticides were used.

### 4.3. Soil Sampling and Analysis

Soil samples were collected pre-harvest (April) and post-harvest (July) in both 2023 and 2024 to assess seasonal changes in soil properties. Five cores (0–20 cm depth) were taken per bed and composited into one sample per plot. Samples were air-dried (<40 °C), sieved to 2 mm, and analyzed for the following parameters:

Soil organic matter (SOM, %): loss-on-ignition at 550 °C for 4 h.

Organic carbon (OC): dry combustion (ECO TruMac CNS analyzer (LECO Corporation, St. Joseph, MI, USA)).

Total nitrogen (Total N): dry combustion (ECO TruMac CNS analyzer (LECO Corporation, St. Joseph, MI, USA)).

Organic nitrogen (Org. N): calculated as Total N—(NH_4_^+^–N + NO_3_^−^–N).

CO_2_–C (ppm): 24 h flush after rewetting air-dried soil, measured with infrared gas analysis [16].

Microbially Active Carbon (MAC, %): calculated as (CO_2_–C ÷ OC) × 100 [17].

We acknowledge the loss-on-ignition method potentially overestimating SOM in high-pH soils. This method was selected for comparability with regional benchmarks; however, we note that future studies will include cross-validation using dry combustion to enhance accuracy.

### 4.4. Soil Infiltration, Moisture, and Temperature

Infiltration: Measured in May of each year using the USDA–NRCS single-ring infiltrometer method [18]. A metal ring (15.2 cm diameter, 13.5 cm height) was inserted 6.5 cm into the soil. A 444 mL water volume (equivalent to 5 cm rainfall) was applied, and the time to complete infiltration was recorded. A second application was made at the same location to determine steady-state infiltration.

Soil moisture and temperature: Recorded weekly at 10 cm depth using an Acclima TDR-315L sensor (Acclima Inc., Meridian, ID, USA). Irrigation was initiated when volumetric water content declined below 21% and continued until near field capacity (≈22–23%). Three readings per plot were averaged and aggregated into monthly means.

### 4.5. Cover Crop, Living Mulch, and Weed Biomass

Aboveground biomass was sampled at six time-points:

July 2022 (peak cover growth), August 2022 (post-termination), March 2023 (start of harvest), July 2023 (end of harvest), March 2024 (start of second harvest), and July 2024 (end of second harvest).

At each date, three 0.3 × 0.3 m quadrats per plot were harvested, separating residue and weed fractions, dried to constant mass, and expressed as t·ha^−1^. After sampling, all plots were hand-weeded to standardize subsequent intervals.

### 4.6. Strawberry Yield

Marketable strawberry yield was measured twice weekly during harvest (April–July) in 2023 and 2024. Fruit were graded as marketable if ripe, defect-free, and of commercial size. Fresh weight was determined using a digital balance (Ohaus Scout SPX, Ohaus Corporation, Parsippany, NJ, USA; precision ± 0.1 g). Plot yields were recorded in grams and converted to t·ha^−1^ using total plot area (beds + furrows). Monthly and cumulative seasonal yields were calculated.

### 4.7. Statistical Analysis

Data were analyzed by one-way ANOVA (CoStat v6.45, CoHort Software, Monterey, CA, USA). to evaluate treatment effects on soil properties, biomass, weed suppression, and yield. Each treatment had four replicates (n = 4). Data were checked for normality and homogeneity of variance (Bartlett’s test). Where treatment effects were significant (*p* ≤ 0.05), means were separated using Tukey’s Honestly Significant Difference (HSD) test. Results are reported as means ± standard deviation (SD), and figure error bars represent standard error of the mean (SE).

## 5. Conclusions

Across two consecutive production seasons in coastal California, rolled cover crop residues—particularly sorghum–pea systems—provided comparable or superior yields to polyethylene mulch while moderating soil temperature, maintaining soil moisture, and improving infiltration. Although soil health responses were modest and primarily seasonal, they aligned with early indicators of improvement typically observed in organic systems.

Weed suppression outcomes varied among treatments. The white clover living mulch achieved the most consistent weed suppression but reduced yield due to in-row competition. Sorghum residues, in contrast, provided a balanced combination of weed control and productivity, while buckwheat offered rapid early-season coverage but decomposed quickly, requiring additional management to sustain ground cover.

The findings confirm that high-residue cover crops can serve as viable alternatives to plastic mulch, reducing environmental impact while maintaining strawberry yields. However, optimizing these systems will require fine-tuning of termination timing, residue management, and mixed-species cover combinations to ensure uniform ground cover and minimize labor costs.

Continued multi-year research should also focus on long-term soil biological responses and operational feasibility under commercial conditions. The transition away from plastic mulch not only supports soil and ecosystem health but also addresses an urgent sustainability challenge associated with microplastic contamination in agricultural systems.

## Figures and Tables

**Figure 1 plants-14-03385-f001:**
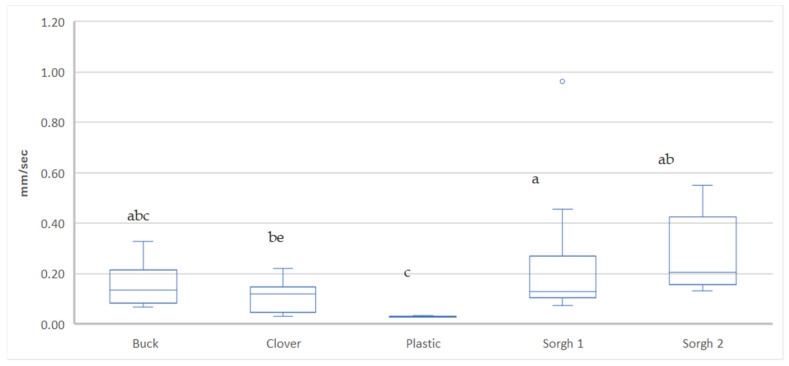
Infiltration Time Across Mulching Treatments in Strawberry Fields. Box-and-whisker plot of soil infiltration rate (mm s^−1^) measured across five mulching treatments in an organic strawberry field in Camarillo, CA, USA: Treatments: Buck = roller-crimped buckwheat/pea mixture (*Fagopyrum esculentum*/*Pisum sativum*); Clover = white clover living mulch (*Trifolium repens*); Sorghum 1 and Sorghum 2 = roller-crimped sorghum–pea mixtures (*Sorghum bicolor* × *S. sudanense*/*P. sativum*); lower and higher sorghum seeding rates, respectively); Plastic = polyethylene mulch. Treatments not sharing the same letter differ significantly according to the Student–Newman–Keuls post hoc test (*p* < 0.05). Boxes represent the interquartile range (IQR), the horizontal line indicates the median, whiskers extend to 1.5× IQR, and circles indicate outliers.

**Figure 2 plants-14-03385-f002:**
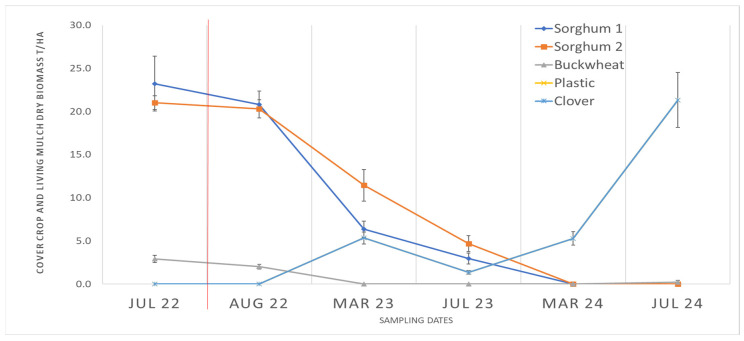
Temporal trends in aboveground cover crop and living mulch biomass (t ha^−1^) under different soil management systems in an organic strawberry field (Camarillo, CA, USA). Treatments: Sorghum 1 = sorghum–sudangrass (*Sorghum bicolor × S. sudanense*) and field pea (*Pisum sativum* subsp. *arvense*) mixture (100 and 30 kg ha^−1^, respectively); Sorghum 2 = sorghum–field pea mixture (200 and 30 kg ha^−1^, respectively); Buckwheat = buckwheat (*Fagopyrum esculentum*) and field pea mixture (67 and 30 kg ha^−1^, respectively); Clover = white clover (*Trifolium repens*) living mulch seeded at strawberry transplanting; Plastic = black polyethylene mulch (control). Weed biomass represents aboveground dry matter averaged over four replicates. Error bars indicate the standard error of the mean (n = 12). Measurements were taken in July 2022 (peak cover crop growth prior to crimping), August 2022 (post-crimping), March 2023 (start of harvest), July 2023 (end of harvest), March 2024 (start of second harvest), and July 2024 (end of second harvest). The red vertical line indicates the timing of cover crop termination.

**Figure 3 plants-14-03385-f003:**
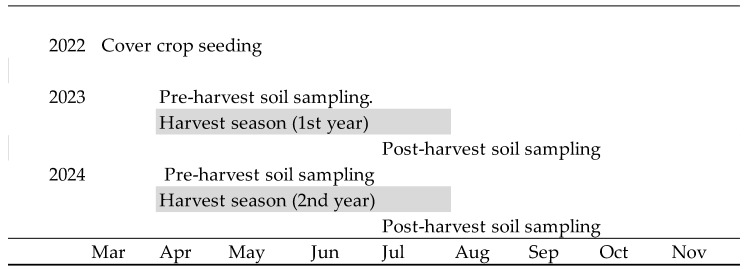
Timeline of cropping system activities in the organic strawberry experiment (2022–2024). Cover crop establishment (May 2022) followed by roller-crimp termination and strawberry transplanting (August 2022). Two consecutive production cycles: 2023 and 2024, each including pre-harvest soil sampling (April), harvest season (April–July), and post-harvest soil sampling (July).

**Table 1 plants-14-03385-t001:** Temporal trends in soil chemical and biological properties under different mulching systems in an organic strawberry field (Camarillo, CA, USA), 2023. Treatments: Buckwheat = roller-crimped buckwheat/pea mixture (*Fagopyrum esculentum × Pisum sativum*); Clover = white clover living mulch (*Trifolium repens*); Sorghum 1 and Sorghum 2 = roller-crimped sorghum–pea mixtures (*Sorghum bicolor × S. sudanense × Pisum sativum*); lower and higher sorghum seeding rates, respectively); Plastic = polyethylene mulch. Soil samples were collected pre (**A**) and post (**B**) harvest. Means ± SD followed by different letters within a column indicate significant differences according to Tukey’s HSD test (*p* ≤ 0.05). Significance codes: ** = *p* ≤ 0.01; * = *p* ≤ 0.05; n.s. = not significant.

(**A**)						
Treatment	SOM (%)	OC (ppm C)	Total N (ppm N)	Org N (ppm N)	CO_2_–C (ppm C)	MAC (%)
Buckwheat	1.95 ± 0.13	109.5 ± 14.4 b	18.15 ± 1.55 b	12.68 ± 2.56 b	39.9 ± 4.8	37.05 ± 7.58
Clover	1.93 ± 0.05	126.0 ± 16.4 ab	16.20 ± 1.21 ab	12.98 ± 1.40 b	44.6 ± 10.9	35.28 ± 5.76
Plastic	1.85 ± 0.10	107.0 ± 6.2 b	23.70 ± 11.81 b	15.33 ± 4.21 ab	39.1 ± 11.2	36.25 ± 9.36
Sorghum 1	2.03 ± 0.10	152.8 ± 22.3 a	22.53 ± 2.40 a	18.83 ± 2.57 a	45.3 ± 11.3	30.93 ± 11.69
Sorghum 2	2.00 ± 0.08	137.8 ± 20.3 ab	19.88 ± 1.96 ab	15.80 ± 1.57 ab	43.9 ± 11.2	31.68 ± 5.77
Sig.	n.s.	**	n.s.	*	n.s.	n.s.
(**B**)						
Treatment	SOM (%)	OC (ppm C)	Total N (ppm N)	Org N (ppm N)	CO_2_–C (ppm C)	MAC (%)
Buckwheat	2.00 ± 0.12	105.3 ± 5.1	33.5 ± 3.6	20.23 ± 0.93	10.50 ± 0.56	31.98 ± 4.56
Clover	2.03 ± 0.10	108.5 ± 7.4	78.5 ± 49.1	22.45 ± 4.06	8.73 ± 2.70	73.50 ± 47.76
Plastic	1.88 ± 0.10	113.8 ± 7.9	33.4 ± 2.7	26.00 ± 16.16	10.73 ± 6.80	29.38 ± 0.42
Sorghum 1	2.05 ± 0.10	124.0 ± 12.8	99.2 ± 94.0	23.30 ± 6.08	10.90 ± 3.60	76.55 ± 65.56
Sorghum 2	2.08 ± 0.05	111.3 ± 5.7	39.6 ± 8.9	21.98 ± 1.34	10.45 ± 1.80	35.53 ± 6.76
Sig.	n.s.	n.s.	n.s.	n.s.	n.s.	n.s.

**Table 2 plants-14-03385-t002:** Temporal trends in soil chemical and biological properties under different mulching systems in an organic strawberry field (Camarillo, CA, USA), 2024. Treatments: Buckwheat = roller-crimped buckwheat/pea mixture (*Fagopyrum esculentum × Pisum sativum*); Clover = white clover living mulch (*Trifolium repens*); Sorghum 1 and Sorghum 2 = roller-crimped sorghum–pea mixtures (*Sorghum bicolor × S. sudanense × Pisum sativum*); lower and higher sorghum seeding rates, respectively); Plastic = polyethylene mulch. Soil samples were collected pre (**A**) and post (**B**) harvest. Means ± SD followed by different letters within a column indicate significant differences according to Tukey’s HSD test (*p* ≤ 0.05). Significance codes: * = *p* ≤ 0.05; n.s. = not significant.

(**A**)						
Treatment	SOM (%)	OC (ppm C)	Total N (ppm N)	Org N (ppm N)	CO_2_–C (ppm C)	MAC (%)
Buckwheat	1.66 ± 0.07 b	173.5 ± 7.3	20.3 ± 2.3	16.9 ± 1.7	17.5 ± 3.0	35.6 ± 12.1
Clover	1.69 ± 0.06 ab	185.0 ± 7.8	21.6 ± 2.4	17.2 ± 1.3	17.0 ± 2.1	32.3 ± 10.4
Plastic	1.65 ± 0.05 b	165.0 ± 12.0	20.0 ± 2.5	15.8 ± 1.6	16.3 ± 2.4	29.5 ± 8.7
Sorghum 1	1.78 ± 0.07 a	178.0 ± 10.5	21.5 ± 2.8	17.0 ± 1.5	15.8 ± 2.2	28.6 ± 9.4
Sorghum 2	1.74 ± 0.09 ab	175.0 ± 6.3	20.9 ± 2.6	16.9 ± 1.4	16.6 ± 1.8	30.2 ± 9.1
Sig.	*	n.s.	n.s.	n.s.	n.s.	n.s.
(**B**)						
Treatment	SOM (%)	OC (ppm C)	Total N (ppm N)	Org N (ppm N)	CO_2_–C (ppm C)	MAC (%)
Buckwheat	1.95 ± 0.05	122.0 ± 4.2	15.0 ± 1.8	9.6 ± 1.0	9.4 ± 0.9	15.4 ± 2.3
Clover	1.97 ± 0.06	125.0 ± 4.6	18.0 ± 1.9	10.8 ± 1.3	10.3 ± 0.8	20.5 ± 2.4
Plastic	1.99 ± 0.07	129.0 ± 9.8	17.0 ± 2.2	9.8 ± 1.5	10.1 ± 0.7	17.4 ± 2.1
Sorghum 1	2.07 ± 0.09	130.0 ± 3.8	18.5 ± 2.4	10.9 ± 1.7	9.9 ± 1.9	18.9 ± 3.6
Sorghum 2	1.91 ± 0.12	119.0 ± 5.6	15.9 ± 2.1	9.2 ± 1.2	8.8 ± 0.8	16.1 ± 2.9
Sig.	*	n.s.	n.s.	n.s.	n.s.	n.s.

**Table 3 plants-14-03385-t003:** Monthly average soil moisture (volumetric %, *v*/*v*) (**A**) and soil temperature (°C) (**B**) under different soil management systems in an organic strawberry field (Camarillo, CA, USA). Treatments: Buckwheat = roller-crimped buckwheat/pea mixture (*Fagopyrum esculentum × Pisum sativum*); Clover = white clover living mulch (*Trifolium repens*); Sorghum 1 and Sorghum 2 = roller-crimped sorghum–pea mixtures (*Sorghum bicolor × S. sudanense × Pisum sativum)*; lower and higher sorghum seeding rates, respectively); Plastic = polyethylene mulch. Measurements were taken weekly at 10 cm depth, 48 h after irrigation. Monthly means were calculated for March–July in 2023 and 2024. Means followed by different lowercase letters within columns indicate significant differences according to Tukey’s HSD test (*p* ≤ 0.05). Significance codes: *** = *p* ≤ 0.001; n.s. = not significant.

(**A**)						
		Soil moisture %, *v*/*v*				
Year	Treatment	Mar	Apr	May	Jun	Jul
2023	Sorghum 1	29.23 ± 2.03 b	25.55 ± 1.39 b	24.99 ± 2.27 b	23.56 ± 1.74 b	21.26 ± 2.31 b
	Sorghum 2	29.94 ± 2.55 b	25.18 ± 2.18 b	23.79 ± 2.52 b	23.22 ± 1.87 b	19.81 ± 2.07 b
	Buckwheat	28.78 ± 1.30 b	24.33 ± 1.66 b	20.65 ± 2.35 c	22.86 ± 2.61 b	19.11 ± 2.62 b
	Clover	29.77 ± 1.39 b	24.28 ± 1.97 b	25.37 ± 2.27 b	20.50 ± 1.81 c	18.50 ± 1.70 c
	Plastic	32.73 ± 2.20 a	29.64 ± 0.90 a	29.71 ± 1.45 a	29.69 ± 1.54 a	28.04 ± 2.28 a
	Sig.	***	***	***	***	***
2024	Sorghum 1	20.20 ± 1.59 b	25.61 ± 1.74 b	21.58 ± 0.79 b	23.78 ± 2.01 b	21.55 ± 4.05
	Sorghum 2	20.85 ± 1.93 b	25.50 ± 1.58 b	20.80 ± 0.91 bc	23.32 ± 1.21 b	25.20 ± 6.38
	Buckwheat	17.94 ± 1.44 c	23.94 ± 1.09 b	19.99 ± 1.07 c	22.67 ± 2.56 bc	22.81 ± 1.83
	Clover	17.75 ± 2.35 c	23.73 ± 2.48 b	21.01 ± 1.38 bc	20.70 ± 1.86 c	27.44 ± 6.70
	Plastic	24.70 ± 1.35 a	29.23 ± 1.49 a	23.73 ± 1.05 a	30.32 ± 1.57 a	25.15 ± 4.83
	Sig.	***	***	***	***	n.s.
(**B**)						
		Soil Temperature (°C)				
Year	Treatment	Mar	Apr	May	Jun	Jul
2023	Sorghum 1	23.64 ± 0.93 b	32.31 ± 2.34	26.78 ± 0.91	25.02 ± 0.64	32.03 ± 0.12
	Sorghum 2	23.85 ± 0.56 b	32.02 ± 2.23	27.16 ± 0.93	24.85 ± 0.76	32.20 ± 0.21
	Buckwheat	23.29 ± 0.82 b	32.50 ± 2.05	27.27 ± 0.60	25.06 ± 0.52	32.05 ± 0.12
	Clover	23.61 ± 0.82 b	32.59 ± 1.36	26.66 ± 0.79	25.01 ± 0.36	32.13 ± 0.13
	Plastic	25.23 ± 0.32 a	32.00 ± 2.06	26.54 ± 1.07	24.75 ± 0.46	32.14 ± 0.20
	Sig.	***	n.s.	n.s.	n.s.	n.s.
2024	Sorghum 1	22.72 ± 0.39 b	26.35 ± 1.03 bc	22.41 ± 0.81 b	24.98 ± 0.63	27.11 ± 0.26
	Sorghum 2	22.62 ± 0.90 b	26.67 ± 1.23 b	22.23 ± 0.55 b	24.93 ± 0.73	27.31 ± 0.48
	Buckwheat	22.55 ± 1.50 b	25.00 ± 0.81 c	22.01 ± 1.12 b	25.01 ± 0.52	27.05 ± 0.43
	Clover	21.07 ± 1.12 c	25.28 ± 1.63 bc	21.86 ± 1.36 b	25.05 ± 0.39	27.27 ± 0.42
	Plastic	24.50 ± 0.65 a	28.56 ± 1.28 a	24.04 ± 1.15 a	24.87 ± 0.54	26.99 ± 0.36
	Sig.	***	***	***	n.s.	n.s.

**Table 4 plants-14-03385-t004:** Monthly and total strawberry yields (t ha^−1^) under different soil management systems in an organic strawberry field (Camarillo, CA, USA), during the 2023 (**A**) and 2024 (**B**) growing seasons. Treatments: Buckwheat = roller-crimped buckwheat/pea mixture (*Fagopyrum esculentum × Pisum sativum* subsp. *arvense*); Clover = white clover living mulch (*Trifolium repens*); Sorghum 1 and Sorghum 2 = roller-crimped sorghum–pea mixtures (*Sorghum bicolor × S. sudanense × P. sativum* subsp. *Arvense*); lower and higher sorghum seeding rates, respectively); Plastic = black polyethylene mulch (control). Yields represent monthly harvest totals (April–July) and cumulative seasonal yield for each year. Means ± SD followed by different lowercase letters within columns indicate significant differences according to Tukey’s HSD test (*p* ≤ 0.05). Significance codes: *** = *p* ≤ 0.001; ** = *p* ≤ 0.01; n.s. = not significant.

(**A**)					
	Yield (t·ha^−1^)				
Treatment	April	May	June	July	Total
Sorghum 1	1.7 bc	7.4 a	8.0 a	5.1 b	22.2 a
Sorghum 2	2.1 b	8.4 a	8.5 a	5.5 b	24.5 a
Buckwheat	1.2 c	3.1 c	4.7 c	4.3 b	13.3 b
Clover	0.3 d	0.1 d	0.0 d	0.2 c	0.6 c
Plastic	4.2 a	6.0 b	5.7 b	7.7 a	23.6 a
Sig.	***	***	**	***	***
(**B**)					
Treatment	April	May	June	July	Total
Sorghum 1	3.9 a	2.0 a	3.0 a	2.8 a	11.6 a
Sorghum 2	3.5 a	2.9 a	2.9 a	2.7 a	11.1 a
Buckwheat	1.6 b	0.6 b	1.4 b	1.4 b	5.0 b
Clover	0.2 bc	0.1 b	0.8 c	0.08 c	0.4 c
Plastic	0.8 c	0.4 b	0.03 bc	0.5 c	2.4 c
Sig.	***	***	***	***	***

**Table 5 plants-14-03385-t005:** Rainfall data were recorded at the Camarillo Airport Weather Station (USW00023136), and irrigation values reflect supplemental water applied to maintain consistent soil moisture across all treatments.

Month	Precipitation (mm)	Irrigation (mm)	Total Applied (mm)
Nov-22	80	0	80
Dec-22	70	0	70
Jan-23	150	0	150
Feb-23	100	0	100
Mar-23	50	0	50
Apr-23	40	15	55
May-23	40	15	55
Jun-23	40	15	55
Jul-23	0	40	40
Aug-23	0	40	40
Sep-23	0	40	40
Oct-23	30	10	40
Nov-23	100	10	110
Dec-23	80	0	80
Jan-24	150	0	150
Feb-24	100	0	100
Mar-24	90	0	90
Apr-24	50	20	70
May-24	50	20	70
Jun-24	40	20	60
Jul-24	20	50	70
Total	1280	295	1575

## Data Availability

The datasets generated and analyzed during the current study are not publicly available due to ongoing related analyses but are available from the corresponding author on reasonable request.

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
