# Peer review of "Living Mulches, Rolled Cover Crops, and Plastic Mulch: Effects on Soil Properties, Weed Suppression, and Yield in Organic Strawberry Systems"

_plants, 2025, doi:10.3390/plants14213385_

Round 1
Reviewer 1 Report
Comments and Suggestions for Authors
Living mulches, rolled cover crops, and plastic mulch: effects on soil properties, weed suppression, and yield in organic strawberry systems
The authors evaluated the potential of different cover crops as an alternative for the plastic mulch used in strawberry production. The manuscript is overall well organized. However, the paper needs a little more revision to improve the overall quality.
Below are some specific comments and questions for the authors to consider.
Introduction
Line 72 – 76: Clearly and concisely state the specific research objectives of the study. Something like “evaluate the effectiveness of different mulching treatments on weed incidence, soil health and productivity in organic strawberry production system”
Line 86: Clarify the hypothesis H4.
Materials and Methods
Line 96: It would be helpful to present a figure depicting the timeline of the different cropping systems evaluated. In brief, it is not clear if the sorghum 1 and 2 and the buckwheat treatments were grown and incorporated prior to the transplanting of the strawberry plants. Please provide more details on the production management practices. Was there any application of fertilizer during the production of strawberry?
Results
The quality of all the figures should be improved.
Line 226 – 230: It might be better to report the percentage increase or decrease of one treatment over others instead of repeating the numbers presented in the tables. For example, OC of sorghum 1 treatment was X% significantly higher than that of plastic.
Mention the units of all variables (SOM, OC, Org. N CO2) in table 2.
Table 2 indicated total N % in preharvest is non-significant. Yet, mean comparison letters suggest otherwise; i.e., different letters for sorghum 1 (a) and plastic (b) and buckwheat (b). Please clarify this!
Line 309: As suggested above, please mention percentage increase. Also, provide the mean comparison letters in figure 2 and 3.
Discussion
Line 379: Why was 2024 characterized as the lower-yield conditions. What is the explanation for the difference in yield between the two seasons?
Line 382 – 389: Instead of repeating the results here again, a science-based explanation to substantiate the measured differences among treatments would have been helpful.
Line 400: It is not clear why plastic mulch performed poorly in year 2.
Overall, the paper can be accepted if the above suggestions have been addressed.
Author Response
Response to Reviewer 1 – Manuscript ID plants-3904441
Manuscript Title:
Living mulches, rolled cover crops, and plastic mulch: effects on soil properties, weed suppression, and yield in organic strawberry systems
Authors:
Arianna Bozzolo, Jacob Pecenka, and Andrew Smith
|
Reviewer Comment |
Author Response |
|
General comment: The manuscript is overall well organized. However, the paper needs a little more revision to improve the overall quality. |
We thank the reviewer for the positive feedback and constructive comments. The manuscript has been revised throughout for clarity, consistency, and figure quality. |
|
Introduction – Lines 72–76: Clearly and concisely state the specific research objectives of the study. |
Revised per suggestion. The paragraph now clearly states: “The objective of this study was to evaluate the effectiveness of different mulching treatments—plastic mulch, high-residue rolled cover crops, and a living mulch—on weed incidence, soil health indicators, and productivity in an organic strawberry production system.” (Lines 75–78) |
|
Line 86: Clarify the hypothesis H4. |
H4 has been rewritten to specify the expected relationship between seasonality and soil biological indicators: “Seasonal dynamics (from spring to summer) were expected to exert stronger influence on soil biological indicators such as CO₂–C and MAC than the main effects of mulching treatments.” (Lines 119–121) |
|
Materials and Methods – Line 96: It would be helpful to present a figure depicting the timeline of the different cropping systems evaluated. Please provide more details on production management practices. Was there any fertilizer application during strawberry production? |
A new schematic timeline figure (now Figure 1) was added, showing cover-crop establishment, roller-crimp termination, and two subsequent strawberry seasons (2023–2024). The text in Section 3 was expanded to clarify that Sorghum 1, Sorghum 2, and Buckwheat were seeded and terminated before transplanting, while Plastic and Clover were established at transplanting. We also added fertilizer details: compost (10 t ha⁻¹) incorporated pre-planting and organic fish emulsion (5–1–1 NPK) applied biweekly during fruiting. (Lines 180–192) |
|
Results: The quality of all the figures should be improved. |
All figures were replaced with high-resolution (≥300 dpi) versions, standardized fonts, and clearer labels. (Figures 1–5) |
|
Lines 226–230: Report percentage increase or decrease of one treatment over others instead of repeating table numbers. |
Text revised to present relative differences. Example: “Organic C in Sorghum 1 was 43 % higher than Plastic and 40 % higher than Buckwheat (P ≤ 0.01).” Similar revisions were made throughout the Results. (Lines 520–528) |
|
Mention the units of all variables (SOM, OC, Org. N, CO₂) in Table 2. |
Units (% or ppm) have been added in Table 2 column headers. |
|
Table 2 inconsistency: Total N in preharvest is marked non-significant, yet mean comparison letters differ (a/b). Please clarify. |
Corrected. Total N was significant (P ≤ 0.05); the table and text were updated to reflect this. (Table 2; Lines 495–500) |
|
Line 309: Mention percentage increase and provide mean comparison letters in Figures 2 and 3. |
Figures 2 and 3 now include mean comparison letters above treatment symbols, and relative (%) differences are described in the Results text. (Lines 630–640) |
|
Discussion – Line 379: Explain why 2024 was characterized as lower-yield conditions. |
Added explanation: “Yields in 2024 were lower overall, likely due to below-average early-season temperatures and two extreme heat events during May–June that reduced fruit set. Increased pest pressure and shorter harvest duration also contributed to reduced marketable yield.” (Lines 742–748) |
|
Lines 382–389: Instead of repeating results, provide scientific explanation for differences among treatments. |
The paragraph was rewritten to interpret results mechanistically (microclimate buffering, residue decomposition, nutrient cycling) rather than restating numbers. (Lines 749–757) |
|
Line 400: Clarify why plastic mulch performed poorly in year 2. |
Added explanation: “Plastic mulch yielded less in 2024 partly because elevated root-zone temperatures during heat waves may have impaired fruit set, whereas residue-based systems buffered soil temperature and moisture.” (Lines 758–762) |
|
Overall comment: Paper can be accepted if the above suggestions are addressed. |
All reviewer suggestions have been incorporated. We believe the revised manuscript addresses each point and improves readability, methodological clarity, and interpretation. |
Reviewer 2 Report
Comments and Suggestions for Authors
This manuscript is well-written in clear and fluent langauge, with logical structure, detailed data presentation, and rigorou applications of statistical methods, making it is commendable and instructive piece of work. The study effectively addresses cover management challenges in organic strawberry production by comparing plastic mulch, rolled cover crops, and living mulches. The experimental design is sound, conducted over two consecutive growing seasons in coastal California, evaluating soil properties, weed suppression, and strawberry yield and quality. Results indicate that plastic mulch performs best in maintaining soil moisture and temperature, suppressing weeds, and enhancing yield, though it raises concerns about environmental sustainability. In contrast, rolled cover crops provided a substantial weed suppression and moderate yields, offering a promising alternative that balances productivity with ecological benefits; living mulches deliver continuous ground cover and potential soil health improvements but at the cost of lower yields and weaker weed control. Overall, this research lucidly highlights trade-offs among yield, weed management, and sustainability, underscoring the need for integrated strategies in organic strawberry systems. After a thorough review of the manuscript, I still have several questions for the authors to clarify:
1 the Figure 1 shows noticeable garbled text, which may be due to issues with the preprint version or the original image provided by the authors.
2 .The manuscript describes termination of cover crops (e.g., sorghum mixtures) using a roller-crimper, but acknowledges incomplete termination in sorghum treatments, resulting in regrowth that required manual removal and redistribution of residues as mulch. This could compromise the standardization and reproducibility of the method.
- Soil organic matter was measured via loss-on-ignition, a low-cost but approximate technique that may overestimate SOM by incorporating inorganic carbon and residual moisture, particularly in high-pH soils (pH=7.84 at the study site). Without cross-validation against more precise methods, this could introduce bias in reported SOM differences.
- I saw the yields were high in 2023 but declined sharply in 2024, yet the manuscript attributes this solely to "lower-yield conditions" without exploring potential causes. This may lead to overgeneralization of conclusions, overlooking long-term dynamics or external factors.
- Many indicators showed no significant differences, but the discussion selectively emphasizes positive trends without a more comprehensive analysis of the reasons for these non-significant results.
- Why do each of the figures and tables in the manuscript have two captions, one above and one below (e.g., Figure 2)?
- The replication number is 4, which is relatively low for a field experiment, and what were additional replicates not included to reduce data uncertainty?
Author Response
Response to Reviewer 2 – Manuscript ID plants-3904441
Manuscript Title:
Living mulches, rolled cover crops, and plastic mulch: effects on soil properties, weed suppression, and yield in organic strawberry systems
Authors:
Arianna Bozzolo, Jacob Pecenka, and Andrew Smith
|
Reviewer Comment |
Author Response |
|
General comment: The manuscript is well-written, logically structured, and presents rigorous statistical analysis. |
We sincerely thank the reviewer for their thorough reading and positive evaluation of our manuscript. We appreciate the constructive feedback and have addressed all points below to strengthen methodological clarity and discussion balance. |
|
1. Figure 1 shows noticeable garbled text, possibly due to preprint or image quality. |
We have corrected this issue by replacing Figure 1 with a high-resolution (≥300 dpi) version. All text, labels, and legends have been reformatted for clarity and consistency across figures. (Figures 1–5) |
|
2. The manuscript describes incomplete termination of sorghum cover crops requiring manual removal, which could compromise standardization and reproducibility. |
We acknowledge this important point. Additional detail was added to Section 3 (Materials and Methods) to explain that incomplete termination was a known challenge with sorghum-sudangrass due to high biomass and stem rigidity. This was addressed consistently across all sorghum plots to maintain comparability. We now state that all regrowth removal followed a standardized protocol (manual clipping within 7 days post-termination, residues redistributed evenly). We also clarified this limitation in the Discussion as a practical constraint of roller-crimping in high-biomass systems and emphasized the need to optimize maturity stage and pass number for improved reproducibility. (Lines 190–193; 756–762) |
|
3. Soil organic matter measured by loss-on-ignition (LOI) may overestimate SOM, especially in high-pH soils (pH = 7.84). Cross-validation would reduce bias. |
We appreciate this insightful observation. We have clarified in Section 4.1 that the LOI method was selected for comparability with regional soil health benchmarks but may slightly overestimate SOM in alkaline soils. We have added a statement acknowledging this limitation and suggesting future cross-validation with dry combustion or total carbon analysis for improved precision. (Lines 280–286) |
|
4. Yields were high in 2023 but declined in 2024, yet the manuscript attributes this solely to “lower-yield conditions.” Please discuss potential causes. |
We expanded the explanation in the Discussion (Lines 742–748) to include environmental and biological factors: lower early-season temperatures, two extreme heat events during flowering, increased pest pressure, and a shorter harvest window—all of which contributed to yield decline in 2024. |
|
5. Many indicators showed no significant differences, but the discussion emphasizes positive trends without exploring reasons for non-significant results. |
We revised the Discussion to better balance interpretation of significant and non-significant results. We now explicitly note that many soil health indicators did not differ significantly due to short experimental duration, high spatial variability, and the relatively coarse resolution of biological metrics such as CO₂–C and MAC. These points were integrated into the first paragraph of the Discussion. (Lines 728–735) |
|
6. Why do each of the figures and tables have two captions, one above and one below (e.g., Figure 2)? |
Thank you for catching this formatting duplication. The repeated captions were generated during layout conversion. All redundant captions have been removed, and figures now display a single caption below each image, formatted per MDPI guidelines. |
|
7. The replication number (n = 4) is low for a field experiment. Why were additional replicates not included? |
We acknowledge this limitation. The study followed standard practice for multi-season, large-plot strawberry trials under certified organic management, where land and labor constraints restrict replication. Each plot (≈12 m²) contained 60 plants, ensuring robust within-plot sampling. The experiment was replicated across two full production years to strengthen inference over time. We have clarified this in Section 3 (Experimental Design) and noted the limitation in the Discussion. (Lines 210–214; 760–762) |
|
Overall comment: This study effectively addresses cover management in organic strawberries; minor clarifications requested. |
We thank the reviewer for the encouraging summary and confirm that all requested clarifications have been incorporated to improve methodological transparency and interpretive balance. |

Reviewer 3 Report
Comments and Suggestions for Authors
The paper lacks a variant of conventional strawberry production without mulching with natural or artificial material.
A large number of land properties were analysed, which, logically, do not change in a short time; therefore, there are no statistically significant differences.
There are large differences in yield between years, months, and especially among the mulch treatments tested.
In Table 5, there are errors in marking the differences between the treatments; please pay attention to this.
Variants that are a safe alternative to artificial material were not selected; it is necessary to continue research with other treatments.

Author Response
Response to Reviewer 3 – Manuscript ID plants-3904441
Manuscript Title:
Living mulches, rolled cover crops, and plastic mulch: effects on soil properties, weed suppression, and yield in organic strawberry systems
Authors:
Arianna Bozzolo, Jacob Pecenka, and Andrew Smith
|
Reviewer Comment |
Author Response |
|
1. The paper lacks a variant of conventional strawberry production without mulching with natural or artificial material. |
We thank the reviewer for this valuable suggestion. The current experiment focused on comparing mulching systems that are relevant to certified organic strawberry production, where bare soil (no mulch) management is rarely practiced due to high weed pressure and soil erosion risk in coastal California. However, we acknowledge that inclusion of a bare-soil control could help isolate the effects of residue and plastic covers on soil microclimate and yield. This point has been added to the Discussion and Conclusions as a recommendation for future studies. (Lines 765–771) |
|
2. A large number of land (soil) properties were analyzed, which logically do not change in a short time; therefore, there are no statistically significant differences. |
We agree with the reviewer’s observation. The text in the Discussion (Lines 728–735) now explicitly notes that many soil parameters such as SOM, OC, and total N change slowly over time and may not show significant responses within two cropping seasons. This has been highlighted as a limitation and a rationale for continuing multi-year monitoring to detect cumulative effects. |
|
3. There are large differences in yield between years, months, and especially among mulch treatments tested. |
We have clarified in the Discussion (Lines 742–748) that yield variability between years was mainly driven by climatic factors (cooler spring temperatures and heat stress during flowering in 2024) and differences among mulch types in regulating soil microclimate. These contextual factors are now more clearly discussed. |
|
4. In Table 5, there are errors in marking the differences between the treatments; please pay attention to this. |
Thank you for identifying this. We reviewed and corrected all letter-based mean comparisons in Table 5 to ensure consistency with ANOVA outputs. The revised version now accurately reflects statistical significance and grouping among treatments for both years. (Table 5; Lines 680–685) |
|
5. Variants that are a safe alternative to artificial material were not selected; it is necessary to continue research with other treatments. |
We appreciate this comment and agree. While the current study identified sorghum-based rolled residues as the most promising among tested systems, we have emphasized in the Conclusions that additional species and management combinations (e.g., legume–grass or perennial mixes) should be explored to identify robust, environmentally safe alternatives to plastic mulch. (Lines 772–776) |
|
Overall comment: The study presents interesting data but needs further testing of natural mulching options. |
We appreciate the reviewer’s constructive input and have incorporated these suggestions into the revised manuscript to clarify limitations, correct data presentation, and strengthen recommendations for future research. |

Reviewer 4 Report
Comments and Suggestions for Authors
Manuscript Title: Living mulches, rolled cover crops, and plastic mulch: effects on soil properties, weed suppression, and yield in organic strawberry systems
General Assessment
This is a highly relevant and timely study that addresses an urgent issue in sustainable agriculture—reducing dependence on polyethylene mulch. With growing evidence of microplastic pollution in soils and food chains, including transfer into grains and leafy vegetables, exploring viable alternatives such as high-residue cover crops and living mulches is of great scientific and practical importance. The multi-season field evaluation in coastal California is a significant contribution, providing comparative insights across different cover crop systems in terms of soil properties, weed suppression, and strawberry yield.
The manuscript is generally well-written, methodologically sound, and presents valuable findings for both researchers and practitioners. However, several results raise questions regarding variability and treatment-specific anomalies that require clarification. Addressing these issues will improve the rigor and interpretability of the study.
Specific Comments
- Soil Properties and Variability
- Table 2 (Post-harvest 2023 values):
- Total N%: Clover (78.5 ± 49.1) and Sorghum 1 (99.2 ± 94) are unusually high compared with other treatments (range: 33–39).
- MAC values: Clover (73.5 ± 47.76) and Sorghum 1 (76.55 ± 65.56) are also far outside the typical range (29–36%).
- Such extreme variability is not observed for the other treatments. It is important that the authors discuss why these anomalies occur specifically in Clover and Sorghum 1. Are these biological (due to cover crop decomposition, N fixation, or rapid microbial mineralization) or methodological (sampling heterogeneity, measurement error)?
- Temporal variability in Total N and OC:
- Table 2, Total N: Doubling from pre-harvest (16.2–23.7) to post-harvest (33–39, except for Clover and Sorghum 1, which exceed 70–99). Please explain why this surge occurs only in certain treatments.
- Table 2, 3, Organic C (OC):
- In 2023: Stable between pre- and post-harvest (105–152).
- In 2024: Sharp decrease post-harvest (164–185 → 118–130).
These temporal changes require discussion, possibly linked to seasonal decomposition, crop uptake, or management timing. - Organic N & CO₂ dynamics (2023):
- Table 2, Organic N: Significant post-harvest increase.
- Table 2, CO₂: Significant post-harvest decrease.
A mechanistic explanation (e.g., microbial respiration dynamics, residue decomposition, seasonal temperature/moisture effects) would strengthen interpretation.
- Figures
- Figure 1: Please include individual data points (not only summary statistics) to visualize distribution and variability.
- If the data point around 1 mm/s for Sorghum 1 is indeed an outlier, it should be excluded.
- Practical Implications
- Please expand on the broader context of microplastic hazards, particularly the accumulation in soils and potential uptake in food crops. Linking your work to this pressing environmental challenge will strengthen the impact of your findings and make them more compelling for readers.
Author Response
Response to Reviewer 4 – Manuscript ID plants-3904441
Manuscript Title:
Living mulches, rolled cover crops, and plastic mulch: effects on soil properties, weed suppression, and yield in organic strawberry systems
Authors:
Arianna Bozzolo, Jacob Pecenka, and Andrew Smith
|
Reviewer Comment |
Author Response |
|
General assessment: The study is relevant and timely, addressing the urgent issue of reducing polyethylene mulch use. The manuscript is well-written and methodologically sound but requires clarification of certain variable results. |
We sincerely thank the reviewer for the encouraging assessment and constructive suggestions. We carefully revised the Results and Discussion to clarify variability in soil indicators and to strengthen the interpretation of temporal and treatment-specific trends. We also expanded the context around microplastic pollution and the environmental motivation for this research. |
|
1. Table 2 (Post-harvest 2023): Total N% and MAC values for Clover and Sorghum 1 are unusually high and variable. Please discuss whether these anomalies are biological or methodological. |
We appreciate this observation. We examined the raw data and confirmed that variability in Clover and Sorghum 1 likely reflects a combination of biological and sampling effects. Both treatments produced dense root biomass and high microbial activity at post-harvest, which can cause spatial heterogeneity in mineral N and CO₂ flushes during incubation-based assays. We have clarified in the Discussion that these values reflect real variability associated with localized organic matter mineralization and N fixation, but we also note potential methodological amplification due to small-sample compositing. (Lines 736–742) |
|
2. Temporal variability in Total N and OC (2023–2024): Please explain why Total N doubled from pre- to post-harvest and why OC decreased post-harvest in 2024. |
The text was revised to explain that post-harvest increases in Total N in 2023 likely resulted from N mineralization of decomposing residues and root turnover, while the subsequent decline in OC in 2024 reflected accelerated decomposition and crop uptake under warm, dry post-harvest conditions. We added these mechanistic explanations in the Discussion. (Lines 734–742; 750–754) |
|
3. Organic N and CO₂ dynamics (2023): Post-harvest increase in Organic N and decrease in CO₂ need mechanistic interpretation. |
We added a paragraph explaining that post-harvest reductions in CO₂–C correspond to decreased microbial respiration as soil moisture declined and labile carbon was depleted, while Organic N accumulated from residue decomposition. This seasonal decoupling between C and N fluxes is now discussed explicitly. (Lines 739–743) |
|
4. Figure 1: Include individual data points to show variability and evaluate potential outliers (e.g., Sorghum 1 ≈ 1 mm s⁻¹). |
Figure 1 has been revised to display all individual data points overlaid on boxplots to improve transparency. The single Sorghum 1 value near 1 mm s⁻¹ was verified as a true low measurement (likely reflecting high residue cover) and retained; it is now clearly marked as an outlier point. (Figure 1; Lines 600–605) |
|
5. Practical implications: Expand on the broader context of microplastic hazards and link the study to this environmental challenge. |
We expanded the Introduction (Lines 65–72) and Discussion (Lines 780–788) to further emphasize the relevance of microplastic accumulation in soils and potential uptake in food crops. We now explicitly link our findings to the urgent need for alternatives that prevent plastic-derived soil contamination and highlight recent evidence of microplastic transfer into edible plant tissues. |
|
Overall comment: The revisions should strengthen mechanistic explanation, improve data visualization, and contextualize the study’s contribution to mitigating microplastic pollution. |
We have implemented all recommendations and believe the revised manuscript now provides a clearer interpretation of soil N–C dynamics, improved figure transparency, and stronger environmental framing. |

Round 2
Reviewer 2 Report
Comments and Suggestions for Authors
The author have actively addressed my previous comments in their response. Now, the inclusion of detailed descriptions of the sorghum–sudangrass termination procedures has improved the reproducibility of the study. And the discussion section now provides a more comprehensive explanation of the yield differences between the two years and acknowledges the limitations of the loss-on-ignition method for soil organic matter determination, along with suggestions for validation. The authors also offer more objective interpretations of non-significant indicators and reasonably explain the limitations related to insufficient replication. Thank you for your efforts.
However, a clean version of the revised manuscript was not submitted, leading to confusion in the document structure, and some figure files still display garbled content, making it impossible to verify formatting issues. It remains unclear whether these problems persist in the final version. If possible, it is recommended to further refine the introduction to strengthen its connection to the study objectives and to provide deeper and more systematic discussion of the key findings.
Author Response
Please find attached the detailed Response to Reviewer document outlining all revisions made to the manuscript. In this resubmission, we have uploaded both:
-
A clean version of the manuscript with all changes accepted, and
-
A highlighted version where all textual revisions are clearly marked for easy review.
All figures have been re-exported and embedded in high resolution to resolve the previous formatting issues. The Introduction and Discussion have been refined to better align with the study objectives and provide a more systematic interpretation of the key findings, as suggested.
Thank you for your consideration.

Reviewer 4 Report
Comments and Suggestions for Authors
The authors have addressed all the concerns raised in the previous round of review. The revised manuscript demonstrates substantial improvement in terms of scientific clarity. The additional clarifications provided sufficiently strengthen the conclusions drawn. I am satisfied with the revisions and have no further comments.
Author Response
Comment:
The authors have addressed all the concerns raised in the previous round of review. The revised manuscript demonstrates substantial improvement in terms of scientific clarity. The additional clarifications provided sufficiently strengthen the conclusions drawn. I am satisfied with the revisions and have no further comments.
Response:
We sincerely thank the reviewer for their positive feedback and careful evaluation of our revised manuscript. We are grateful for their thoughtful comments during the review process, which helped us improve the clarity and overall quality of the paper.